:ọ: PLOS | ONE

# LFRET, a novel rapid assay for anti-tissue transglutaminase antibody detection

Juuso Rusanen[1]*, Anne Toivonen[2,3], Jussi Hepojoki[1,4], Satu Hepojoki[1], Pekka Arikoski[5], Markku Heikkinen[6], Outi Vaarala[7], Jorma Ilonen[8], Klaus Hedman[1,2]

**1** University of Helsinki, Medicum, Department of Virology, Helsinki, Finland, **2** Laboratory Services (HUSLAB), Department of Virology and Immunology, University of Helsinki and Helsinki University Hospital, Helsinki, Finland, **3** Department of Clinical Microbiology, Institute of Clinical Medicine, University of Eastern Finland, Kuopio, Finland, **4** Institute of Veterinary Pathology, Vetsuisse Faculty, University of Zurich, Zurich, Switzerland, **5** Department of Pediatrics, Kuopio University Hospital and University of Eastern Finland, Kuopio, Finland, **6** Department of Gastroenterology, Kuopio University Hospital, Kuopio, Finland, **7** Clinicum, University of Helsinki, Helsinki, Finland, **8** Immunogenetics Laboratory, Institute of Biomedicine, University of Turku and Clinical Microbiology, Turku University Hospital, Turku, Finland

* juuso.rusanen@helsinki.fi

**Data Availability Statement:** All relevant data are within the paper and its Supporting Information files.

## Abstract

The diagnosis of celiac disease (CD) is currently based on serology and intestinal biopsy, with detection of anti-tissue transglutaminase (tTG) IgA antibodies recommended as the first-line test. Emphasizing the increasing importance of serological testing, new guidelines and evidence suggest basing the diagnosis solely on serology without confirmatory biopsy. Enzyme immunoassays (EIAs) are the established approach for anti-tTG antibody detection, with the existing point-of-care (POC) tests lacking sensitivity and/or specificity. Improved POC methods could help reduce the underdiagnosis and diagnostic delay of CD. We have previously developed rapid homogenous immunoassays based on time-resolved Förster resonance energy transfer (TR-FRET), and demonstrated their suitability in sero-diagnostics with hanta- and Zika virus infections as models. In this study, we set out to establish a protein L -based TR-FRET assay (LFRET) for the detection of anti-tTG antibodies. We studied 74 patients with biopsy-confirmed CD and 70 healthy controls, with 1) the new tTG-LFRET assay, and for reference 2) a well-established EIA and 3) an existing commercial POC test. IgG depletion was employed to differentiate between anti-tTG IgA and IgG positivity. The sensitivity and specificity of the first-generation tTG-LFRET POC assay in detection of CD were 87.8% and 94.3%, respectively, in line with those of the reference POC test. The sensitivity and specificity of EIA were 95.9% and 91.9%, respectively. This study demonstrates the applicability of LFRET to serological diagnosis of autoimmune diseases in general and of CD in particular.

## Introduction

The diagnosis of celiac disease (CD) is conventionally based on the combination of serology and duodenal biopsy, with detection of IgA anti-tTG antibodies recommended as the first-line

**Funding:** This work was funded by Special Research Funds for University Hospitals in Finland (to J.I., stm.fi), Päivikki and Sakari Sohlberg Foundation (to J.I., pss-saatio.fi), Sigrid Jusélius Foundation (to J.I. and K.H., sigridjuselius.fi), Medical Society of Finland (FLS) (to K.H., fls.fi), Magnus Ehrnrooth Foundation (to K.H., magnusehrnroothinsaatio.fi), Finnish Society of Sciences and Letters (to K.H., scientiarum.fi), and the Research Funds of University of Helsinki and Helsinki University Hospital (to K.H., hus.fi) and the Academy of Finland (to J.H., grant 1308613, aka. fi). The funders had no role in study design, data collection and analysis, decision to publish, or preparation of the manuscript.

**Competing interests:** Some of the authors are inventors in a patent "Protein L based bioassay method for determining presence of soluble antibodies in a sample and kit therefore (WO2015128548)" owned by University of Helsinki describing the LFRET assay utilized in the manuscript. (https://patentscope.wipo.int/search/en/detail.jsf?docId=WO2015128548). This does not alter our adherence to PLOS ONE policies on sharing data and materials and all authors declare that the work was carried out in accordance with good research ethics.

test [1–3]. Total IgA is measured to avoid false negative results in patients with IgA deficiency [1–4]. Other serological markers of CD include antibodies against endomysium antigen (EMA) and deamidated gliadin peptides (DGP), however, somewhat laborious measuring techniques and subjective interpretation (EMA) or weaker specificity (DGP) hampers their use in diagnostics. Additionally, HLA (human leukocyte antigen) testing may aid in ruling out CD, as almost all patients with CD display HLA-DQ2.5 or -DQ8 [5]. Emphasizing the increasing importance of serology, European guidelines allow the diagnosis of symptomatic children to be based on serological markers only [4]. In fact, recent evidence suggests that serological diagnosis would suffice for adults and asymptomatic children [6, 7].

Enzyme immunoassays (EIA) and point-of-care (POC) tests serve as detection methods for anti-tTG antibodies. EIA, with its high sensitivity and specificity, is the most widespread method. However, it requires dedicated laboratory infrastructure, and the results are available at best within some hours. The majority of POC diagnostics is performed using lateral flow assays (LFA), which unlike EIA are rapid but suffer from lower sensitivity (91% vs. 94%, respectively) and specificity (95% vs. 97%, respectively) in detecting biopsy-confirmed CD [8, 9]. Lacking quantitation, the existing anti-tTG IgA POC tests cannot replace EIAs in the diagnostic algorithm of CD as per the European Society for Paediatric Gastroenterology Hepatology and Nutrition (ESPGHAN) [4]. Also, from the follow-up perspective, a quantitative result would be desirable.

Better POC tests could lower the testing threshold and help reduce the diagnostic delay and underdiagnosis of CD. It is estimated that 83–90% of CD patients remain undiagnosed [10], having a markedly reduced quality of life as compared to those diagnosed and treated [11]. Moreover, delayed diagnosis [12, 13] is associated with persistent symptoms [14] leading to increased use of healthcare services, and a decreased quality of life even after the diagnosis and treatment [15].

TR-FRET (time-resolved Förster resonance energy transfer) is a phenomenon occurring when two fluorophores, donor and acceptor, are in close proximity. Excitation of the donor leads to energy transfer to the acceptor, which then emits the energy at a characteristic wavelength. The TR-FRET efficiency depends inversely on the distance between the two fluorophores. Background autofluorescence is minimized by time-resolved measurement, enabled by chelated lanthanide fluorophores with a long fluorescence half-life. TR-FRET has been employed widely in research and diagnosis to investigate e.g. protein-protein interactions and disease markers [16].

We have previously developed a rapid wash-free TR-FRET -based method for antibody detection, termed protein L FRET assay (LFRET) [17]. LFRET employs a donor-labeled antigen, and an acceptor-labeled protein L that binds the kappa (κ) light chains of all immunoglobulin classes. If the clinical sample contains antibodies against the antigen, they will bring the fluorophores to close proximity. Thus, the TR-FRET signal tells that the sample contains the antibodies of interest. The LFRET signal can be measured without additional steps shortly after combining the sample with the reagent mix, allowing for rapid point-of-care diagnosis. We have provided proof-of-concept for the LFRET assay in serodiagnostics using hanta- and Zika virus infections as models [18, 19].

To achieve a test quicker than EIA and with a higher diagnostic utility compared to LFA, and to demonstrate the applicability of the LFRET approach to autoimmune diagnosis, we set out to establish an LFRET assay for the detection of IgA-class anti-tTG antibodies. Using a panel of serum/plasma samples from patients with biopsy-confirmed CD and healthy controls, we herein demonstrate that tTG-LFRET can indeed be utilized in serological diagnosis of CD with a performance comparable to existing POC tests.

## Materials and methods

### Samples

The study included serum and/or plasma samples of 74 Finnish patients, 43 children and 31 adults, with CD confirmed by a duodenal biopsy showing villous atrophy and crypt hyperplasia corresponding to Marsh classes 3A-C. HLA-DQ2.5 and -DQ8 molecules, encoded by the HLA-DQA1 and DQB1 alleles, were analyzed as described [20, 21]. DQ2.5, DQ8 or both were positive for 59, 5 and 4 patients, respectively. With 6 individuals, the HLA status was not analyzed.

The control group comprised serum and/or plasma samples from 70 healthy individuals, including 47 children and 23 adults. DQ2.5, DQ8 or both were positive for 34, 31 and 4 individuals, respectively.

The study was approved by the Ethics Committee of Kuopio University Hospital and written informed consent was obtained from all subjects (from parents/guardians of all children and the children themselves, if >10 years of age).

### Proteins

Recombinant protein L (Thermo Scientific) was labeled with Alexa Fluor 647 (AF) to yield AF-labeled protein L (AF-L), as described [18]. Europium-labeled tTG (Eu-tTG) was generated by labeling baculovirus/Sf9-expressed tTG (Diarect AG) with QuickAllAssay Eu-chelated protein labeling kit (BN Products and Services) according to the manufacturer's instructions. IgG-free bovine serum albumin (BSA) used in LFRET assay was from Jackson ImmunoResearch Inc.

### tTG-LFRET assay

The LFRET assay principle has been described previously [17, 18], and is illustrated in Fig 1. Unlike in those papers, the results here are given as averages of normalized TR-FRET signal values, not divided by the TR-FRET signal of the negative control. To establish an LFRET assay for anti-tTG antibodies the concentrations of assay components were optimized by cross-titration, using panels of 5 to 15 samples shown anti-tTG-IgA positive or negative by FEIA (fluorescent enzyme immunoassay). The optimal on-plate serum dilution was found to be 1/100, and the optimal on-plate concentrations for AF-L and Eu-tTG, 250 nM and 5 nM, respectively. To determine the incubation time, measurements were done at 0, 7, 15, 22, 30, 45, 60 and 90 minutes after mixing the reagents. TR-FRET signals were measured with Wallac Victor$^2$ fluorometer (PerkinElmer) and normalized as described previously [22].

### IgG depletion

To distinguish between anti-tTG IgA- and IgG-class antibodies, GullSORB (Meridian Bioscience, Inc.) was used to deplete the samples of IgG, as described [18]. All samples were studied by tTG-LFRET with and without IgG depletion.

### Reference methods and statistical analyses

The tTG IgA reference test was EliA Celikey IgA (Thermo Scientific, Phadia GmbH), a FEIA used widely by clinical laboratories. Total IgA was measured with an accredited in-house method of HUSLAB (Hospital District of Helsinki and Uusimaa, Laboratory Services, Finland). If the total IgA measurement was indicative of selective IgA deficiency, tTG IgG was measured by EliA Celikey IgG (Thermo Scientific, Phadia GmbH).

# tTG-LFRET

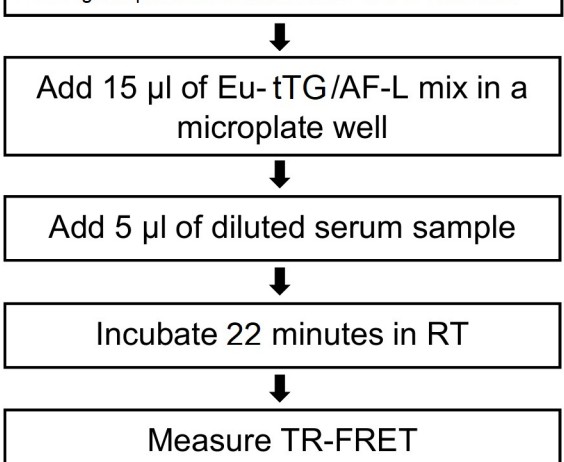

Fig 1. Simplified protocol for tissue transglutaminase protein L TR-FRET assay. Eu-tTG = Europium-labeled tissue transglutaminase, AF-L = Alexa Fluor™ 647 -labeled protein L; TR-FRET = time-resolved Förster resonance energy transfer; RT = room temperature. We used TBS-BSA (50mM Tris-HCl, 150mM NaCl, pH 7.4, 0.2% BSA) for all component dilutions. On-plate serum dilution was 1/100 and reagent concentrations were 250 nM for AF-L and 5 nM for Eu-tTG. For further details, see the previous publication [18].

A commercial rapid lateral flow test (Celiac Disease Quick Test, Biohit) for anti-tTG antibodies performed according to the manufacturer's instructions was used as an additional reference.

All statistical analyses were performed with R version 3.5.1.

## Results

### tTG-LFRET incubation time, cutoff values and performance

An LFRET assay for detection of anti-tTG antibodies (tTG-LFRET) was set up using recombinant Eu-labeled tTG antigen and AF-labeled protein L. The assay conditions were first optimized utilizing 15 selected samples (included in the 144-sample panel) known to be negative (n = 7) or positive (n = 8) for IgA-class anti-tTG antibodies. Then, using the optimized conditions, all 144 samples were tested with tTG-LFRET. While most of the CD patient samples were anti-tTG positive already in the first TR-FRET measurement immediately after mixing the reagents, the best balance between sensitivity, specificity and incubation time was obtained at 22 minutes' assay time (S1 Fig and S1 Table).

To determine the assay cutoff we measured the tTG-LFRET signals for 67 tTG-antibody negative samples, and set the LFRET cutoff at mean plus two standard deviations (SD) (35.438 + 2 × 5.316 = 46.07 counts).

**Table 1. Sensitivity and specificity of LFRET, FEIA and LFA by age group and altogether.**

|  | Group | Subjects | TP | FP | TN | FN | Sensitivity | Specificity |
|---|---|---|---|---|---|---|---|---|
| tTG-LFRET | Children | 90 | 43 | 3 | 44 | 0 | 100.0% | 93.6% |
|  | Adults | 54 | 22 | 1 | 22 | 9 | 71.0% | 95.7% |
|  | **Total** | **144** | **65** | **4** | **66** | **9** | **87.8%** | **94.3%** |
| FEIA | Children | 90 | 43 | 6 | 41 | 0 | 100.0% | 87.2% |
|  | Adults | 54 | 28 | 0 | 23 | 3 | 90.3% | 100.0% |
|  | **Total** | **144** | **71** | **6** | **64** | **3** | **95.9%** | **91.4%** |
| LFA | Children | 90 | 43 | 3 | 44 | 0 | 100.0% | 93.6% |
|  | Adults | 54 | 22 | 1 | 22 | 9 | 71.0% | 95.7% |
|  | **Total** | **144** | **65** | **4** | **66** | **9** | **87.8%** | **94.3%** |

TP = true positive, FP = false positive, TN = true negative, FN = false negative. tTG-LFRET = tissue transglutaminase protein L–based Förster resonance energy transfer assay. FEIA = fluorescent enzyme immunoassay (Phadia EliA Celikey IgA). LFA = Lateral flow assay (Biohit Celiac Disease Quick Test).

We then assessed the diagnostic performance of the tTG-LFRET assay by analyzing altogether 144 serum/plasma samples from 74 CD patients and 70 healthy controls with HLA-associated genetic risk for CD. The sensitivity and specificity of tTG-LFRET in detection of biopsy-proven CD were 87.8% (65/74) and 94.3% (66/70), respectively (Table 1).

## Comparison of tTG-LFRET with FEIA and lateral flow

To compare tTG-LFRET performance with existing assays, we analyzed the above described sample panel using anti-tTG IgA FEIA and a commercial anti-tTG lateral flow assay. By applying the manufacturer-defined cutoffs and considering equivocal results as positive, the sensitivity of anti-tTG FEIA in detection of CD was 95.9% (71/74) and the specificity was 91.4% (64/70) (Table 1). The respective values for the commercial lateral flow assay were 87.8% (65/74) and 94.3% (66/70). The samples incorrectly identified by any of the methods have been listed in Table 2.

Notably, all samples (n = 42) with a high FEIA result (above 70 U/ml) yielded positive tTG-LFRET results (Fig 2). The Pearson correlation between LFRET and FEIA results was 0.85.

## IgG depletion in tTG-LFRET

In an earlier study we employed IgG depletion to distinguish between IgM and IgG in an infectious disease application of LFRET [18]. Here we used the same approach to determine if the sample is anti-tTG IgA-positive or IgA-negative yet IgG-positive. After IgG depletion, using a constant cutoff (mean + 2 × SD, equal to 34.392 + 2 × 5.117 = 44.63 counts), the LFRET assay sensitivity and specificity would be 77.0% (57/74) and 95.7% (67/70), respectively (Fig 3).

The reduction in tTG-LFRET signal (in %) due to IgG depletion was determined for each of the 144 samples. The average reduction plus 2.5 × SD, corresponding to a 59% reduction in tTG-LFRET signal, was chosen as cutoff, with greater reduction taken as indication of IgG-class LFRET positivity. Hence, samples in which upon IgG depletion the tTG-LFRET signal both 1) fell down by >59% and 2) went beyond the cutoff of 44.63 counts, were considered anti-tTG IgG-positive yet IgA-negative. As no such samples were found in the panel, two of them were obtained from HUSLAB to validate the threshold. Upon IgG depletion, the tTG-LFRET signal levels for these samples decreased by 70% and 87% and went below the

**Table 2. Samples identified incorrectly by any of the methods used.**

| Sample | Group | CD status | LFRET (counts, mean of duplicates ± SD) | FEIA (U/ml) | LFA | Incorrectly identified by |
|---|---|---|---|---|---|---|
| 1 | Children | + | 42 ± 8.1 | 12 | + | LFRET |
| 2 | Adults | + | 35 ± 6.1 | 1.4 | - | LFRET, FEIA, LFA |
| 3 | Adults | + | 35 ± 3.8 | 13 | + | LFRET |
| 4 | Adults | + | 36 ± 9.8 | 20 | + | LFRET |
| 5 | Adults | + | 41 ± 9.6 | 15 | + | LFRET |
| 6 | Adults | + | 43 ± 5.5 | 6.3 | - | LFRET, FEIA, LFA |
| 7 | Adults | + | 46 ± 1.7 | 15 | - | LFA |
| 8 | Adults | + | 37 ± 4.3 | 20 | + | LFRET |
| 9 | Adults | + | 34 ± 2.3 | 1 | - | LFRET, FEIA, LFA |
| 10 | Adults | + | 46 ± 1.1 | 13 | - | LFA |
| 11 | Adults | + | 48 ± 12.5 | 14 | - | LFA |
| 12 | Adults | + | 54 ± 4.6 | 33 | - | LFA |
| 13 | Adults | + | 43 ± 0.9 | 15 | - | LFRET, LFA |
| 14 | Adults | + | 42 ± 1.1 | 10 | - | LFRET, LFA |
| 15 | Children | - | 39 ± 5.3 | 10 | - | LFRET, LFA |
| 16 | Children | - | 47 ± 1.6 | 22 | + | LFRET, FEIA, LFA |
| 17 | Children | - | 47 ± 0.8 | 18 | - | LFA |
| 18 | Children | - | 36 ± 1.8 | 14 | + | FEIA, LFA |
| 19 | Children | - | 79 ± 9.9 | 35 | - | LFRET, FEIA |
| 20 | Adults | - | 27 ± 13.8 | 1 | + | LFA |
| 21 | Adults | - | 46 ± 0.3 | 1.7 | - | LFRET |

CD status = celiac disease (CD) status as defined by biopsy (+ = CD,— = no CD). LFRET (protein L TR-FRET assay) / FEIA (fluorescent enzyme immunoassay) / LFA (lateral flow assay) = CD status as suggested by each method, darker background indicating a result suggestive of CD. For LFRET and FEIA, quantitative results are included, as photons for LFRET and as U/ml for FEIA. Cutoffs for LFRET and FEIA are 45 counts and 7 U/ml, respectively.

positivity cutoff; hence the samples were correctly identified as anti-tTG IgA-negative yet IgG-positive (Fig 4).

## Discussion

We established a rapid LFRET assay for the detection of anti-tTG IgA-class antibodies (tTG-LFRET) for serological screening of celiac disease (CD). CD affects 1% of world population and the typical diagnostic delay is 5–10 years [12, 13]. In both developed and resource-poor countries, the vast majority of patients currently remain undiagnosed. Patients with undiagnosed and untreated CD suffer from decreased quality of life and increased risk of various conditions including neuropathy, liver disease as well as osteoporosis [2, 5]. Thus, low-threshold diagnostic testing and screening of risk groups such as first-degree relatives of CD patients is warranted. Reliable anti-tTG POC diagnostics could lower the threshold for CD screening and help overcome the diagnostic delay.

Herein we harnessed LFRET [17], previously applied in infectious disease diagnostics [18, 19], for an autoimmune disease identified by tTG-specific IgA antibodies. This paves way for development of LFRET assays for other autoimmune diseases. While in CD the tTG autoantigen is well-characterized, the less defined antigens in some other autoimmune diseases provide a challenge for POC diagnostics. Nevertheless, with well-defined autoantigens, e.g. ANCA-associated vasculitis, anti-GBM disease, pemphigus and pemphigoid could be suitable targets for LFRET-based assay development.

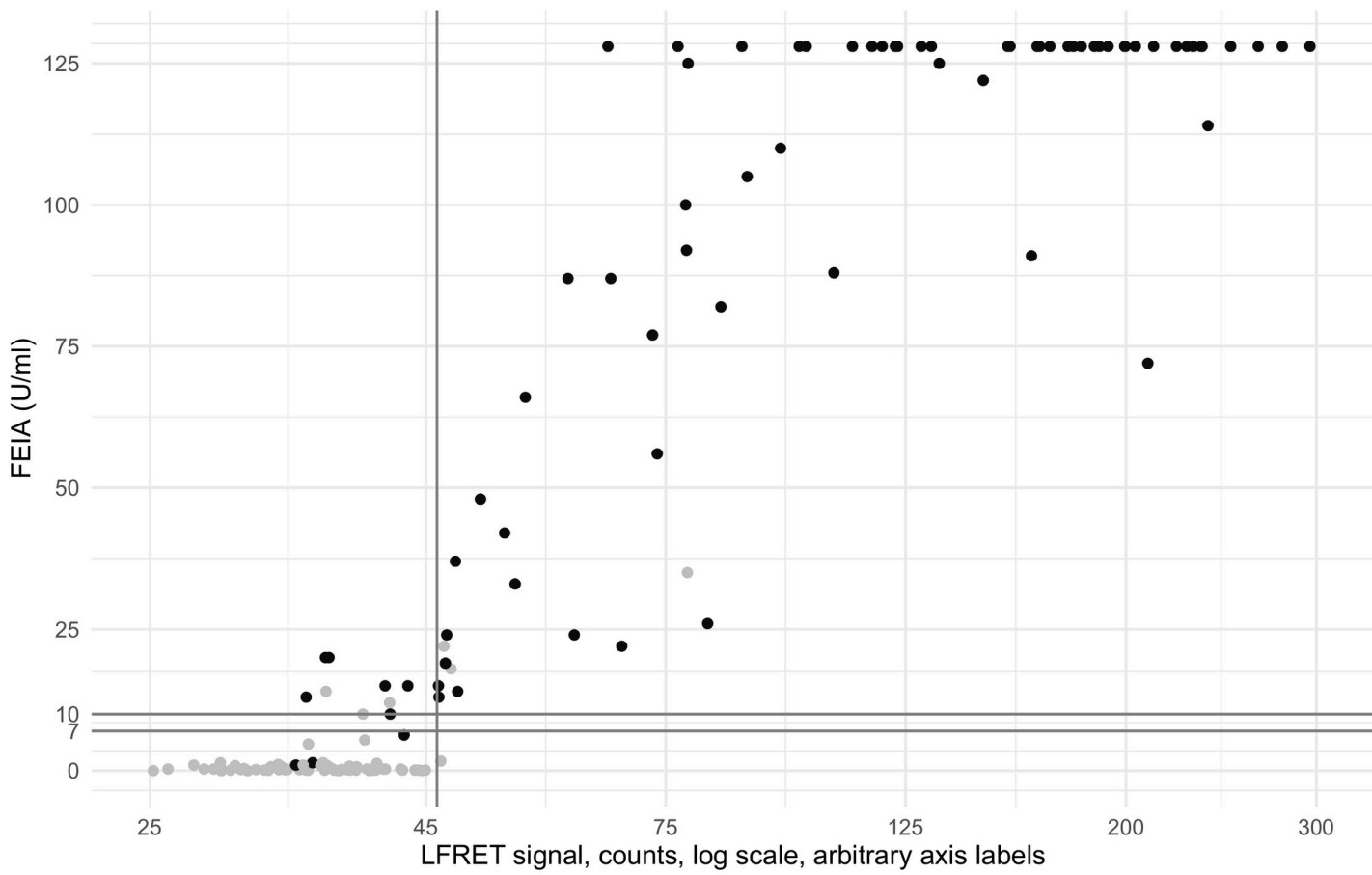

**Fig 2. Anti-tTG-IgA FEIA results (x-axis) compared to LFRET results (y-axis) without IgG depletion.** FEIA = fluorescent enzyme immunoassay. LFRET = protein L–based time-resolved Förster resonance energy transfer assay. FEIA result is expressed as U/ml. LFRET result is expressed as average of normalized acceptor wavelength emission counts from two replicates of the same sample, with two consecutive measurements from both replicates. Patients with biopsy-confirmed celiac disease (CD) are marked with a darker spot. The solid lines indicate cutoffs for FEIA and LFRET, with the area between 7 and 10 U/ml corresponding to an equivocal result for FEIA. FEIA cutoffs are set as determined by the manufacturer. LFRET cutoffs were determined by measuring the tTG-LFRET signals for 67 tTG-antibody negative samples (as defined by FEIA) and setting the LFRET cutoff at the mean LFRET signal plus two standard deviations (SD) ($35.438 + 2 \times 5.316 = 46.07$ counts). Pearson correlation coefficient between FEIA and LFRET results is 0.85.

The sensitivity and specificity of tTG-LFRET for biopsy-confirmed CD were 87.8% and 94.3%, respectively. While the commercial LFA was equal to tTG-LFRET, FEIA exhibited higher sensitivity (95.9%) but lower specificity (91.9%). In line with our LFA results, the existing anti-tTG IgA POC tests have been shown to identify biopsy-confirmed CD with a pooled sensitivity and specificity of 90.5% (95% CI 82.3% –95.1%) and 94.8% (95% CI 92.5% –96.4%), respectively [9]. EIAs have been shown to perform with a slightly lower pooled sensitivity of 93.0% (95% CI 91.2% –94.5%), yet with a higher pooled specificity of 96.5% (95% CI 95.2% – 97.5%) [8]. Interestingly, all of our CD patients anti-tTG negative by FEIA were likewise negative by tTG-LFRET and LFA (Table 2). Lowering the tTG-LFRET cutoff, for enhanced sensitivity of 93.2%, would decrease the specificity to 81.4%. In general, the somewhat lower sensitivity of tTG-LFRET compared to FEIA was confined to samples weakly positive in the latter assay (Fig 2), and might result from the europium label masking some of the tTG epitopes. Another contributing factor could be the variation in immunoglobulin light chain

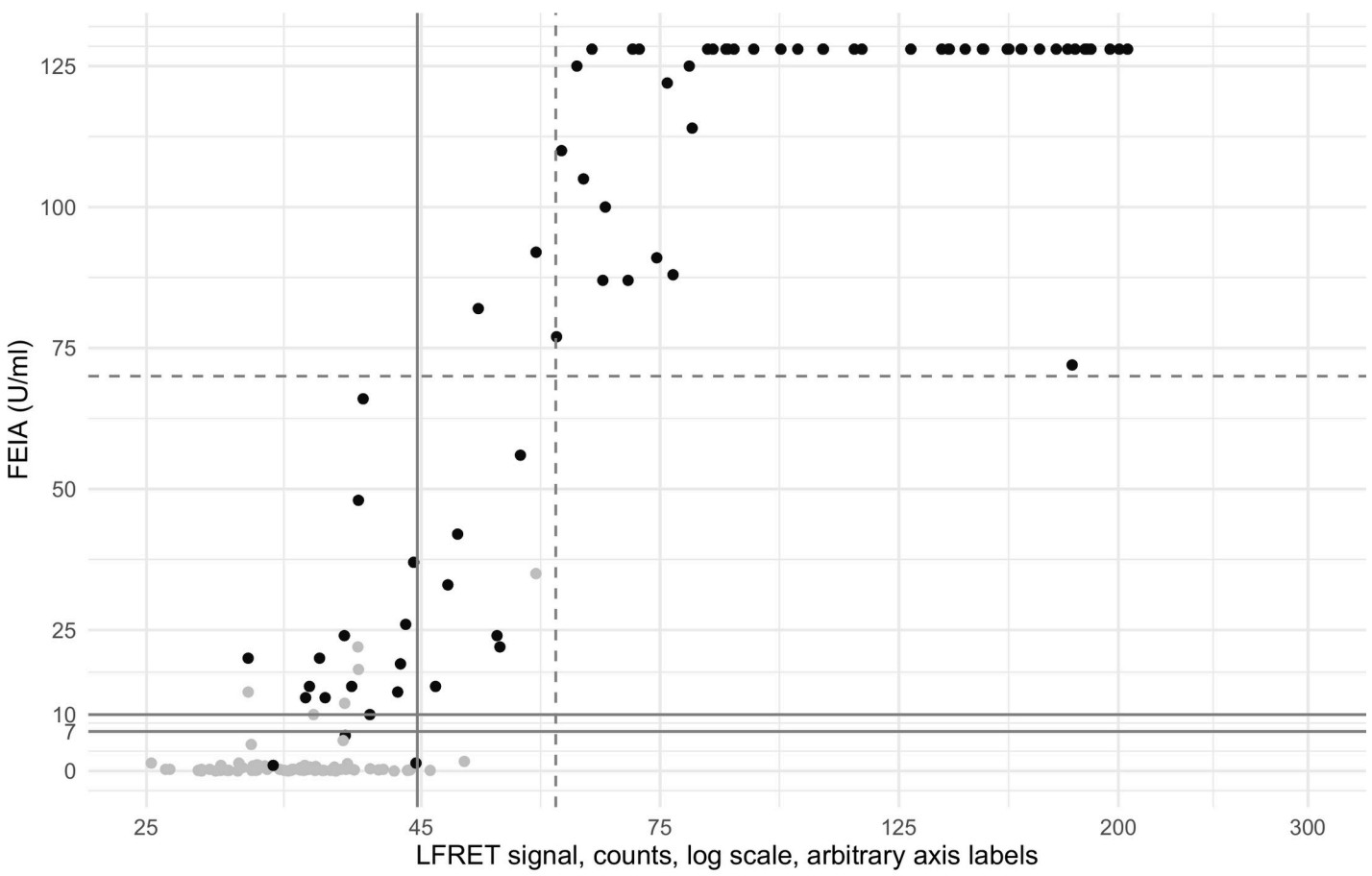

Legend: ● No celiac disease (n=70)  ● Biopsy−confirmed celiac disease (n=74)

**Fig 3. Anti-tTG-IgA FEIA results (x-axis) compared to LFRET results (y-axis) with IgG depletion.** FEIA = fluorescent enzyme immunoassay. LFRET = protein L–based time-resolved Förster resonance energy transfer assay. FEIA result is expressed as U/ml. LFRET result is expressed as average of normalized acceptor wavelength emission counts from two replicates of the same sample, with two consecutive measurements from both replicates. Patients with biopsy-confirmed celiac disease (CD) are marked with a darker spot. The solid lines indicate cutoffs for FEIA and LFRET, with the area between 7 and 10 U/ml corresponding to an equivocal result for FEIA. FEIA cutoffs are set as determined by the manufacturer. LFRET cutoffs were determined by measuring the tTG-LFRET signals for 67 tTG-antibody negative samples (as defined by FEIA) and setting the LFRET cutoff at the mean LFRET signal plus two standard deviations (SD) ($34.392 + 2 \times 5.117 = 44.63$ counts). The dashed line represents x10 upper limit of normal (ULN) for FEIA, and for LFRET a cutoff for detection of samples with a FEIA result above x10 ULN. Pearson correlation coefficient between FEIA and LFRET (with IgG depletion) results is 0.83.

composition: the anti-tTG antibodies in some individuals might mostly have light chains of type λ, nonbinding protein L. Patients with a high-positive FEIA result of >70 U/ml, i.e., >10 times the upper limit of normal (ULN) were all correctly positive in tTG-LFRET, both with and without IgG depletion. This could reflect greater anti-tTG antibody repertoire with antibodies containing both κ and λ light chains. An anti-tTG IgA result of >10x ULN is among the requirements of ESPGHAN criteria for CD diagnosis without biopsy [4]. With IgG depletion, our tTG-LFRET cutoff could be set at 60 counts (Fig 3, dashed lines): A tTG-LFRET result higher than this equals a FEIA result of >10x ULN (> 70 U/ml) with 95% sensitivity (40/42 samples) and 100% specificity (142/142). Indeed, all of the patients exceeding this cutoff had biopsy-confirmed CD. Thus, individuals with an IgG-depleted tTG-LFRET result above 60 can be considered to have CD at a high degree of certainty, possibly without the need for biopsy. Importantly, such patients made up more than a half of those diagnosed with CD in the present study (40/74).

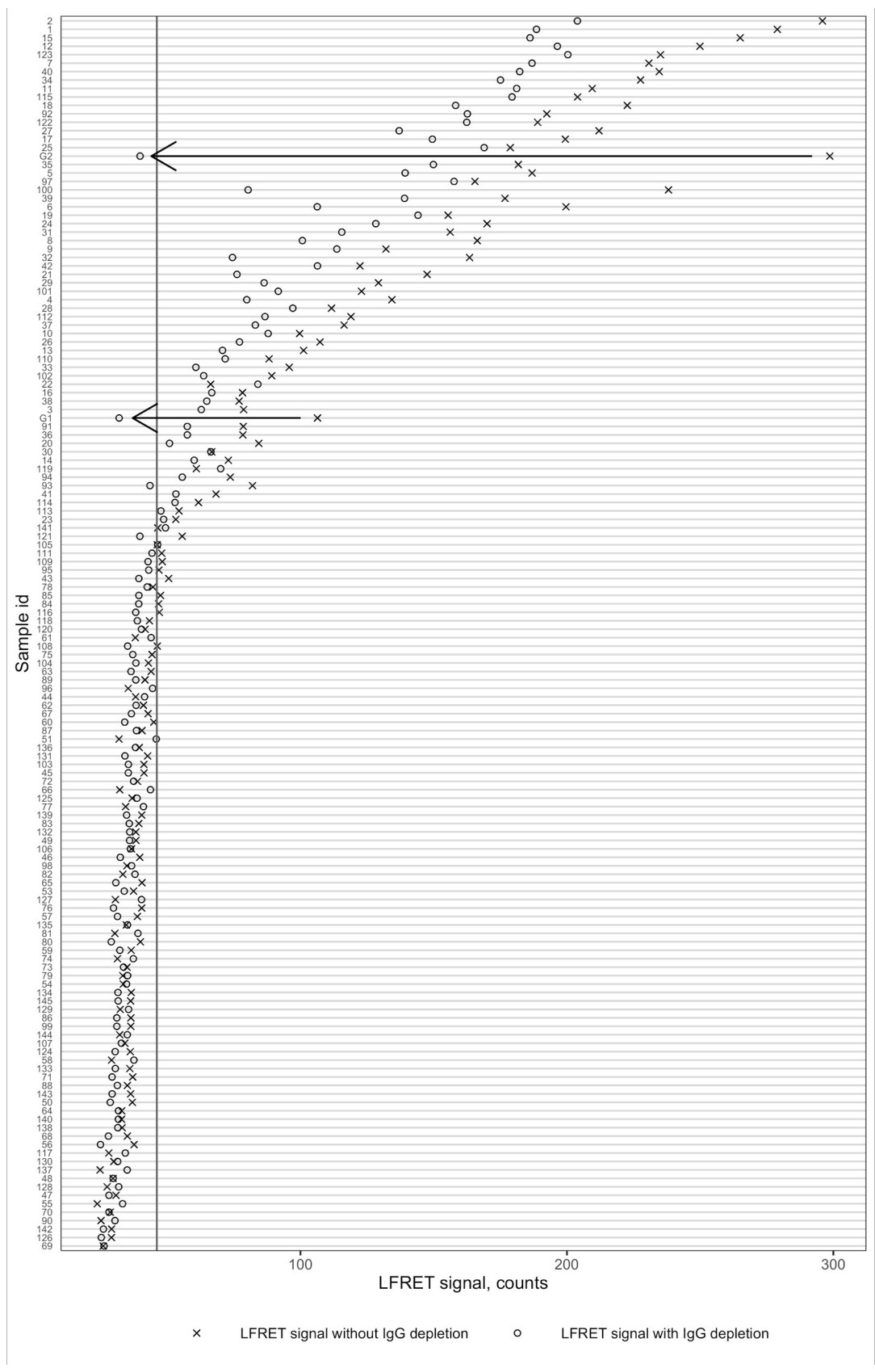

**Fig 4. tTG-LFRET signal change with IgG depletion.** tTG-LFRET = tissue transglutaminase protein L–based time-resolved Förster resonance energy transfer assay. On the x-axis, x marks the LFRET signal without IgG depletion (by GullSORB treatment) and o marks the signal with IgG depletion. On the y-axis are all the studied samples, labels indicating the sample id. An algorithm for differentiation of IgG+/IgA- samples from IgA+ samples was defined as follows: First, the reduction in tTG-LFRET signal (in %) due to IgG depletion was calculated for each of the 144 study samples. The average reduction plus $2.5 \times$ SD, corresponding to a 59% reduction in tTG-LFRET signal, was chosen as cutoff, with greater reduction taken as indication of IgG-class LFRET positivity. Second, to consider the samples IgA-negative, IgG depletion had to reduce the signal below the cutoff as determined by the mean LFRET signal plus two standard deviations (SD) ($34.392 + 2 \times 5.117 = 44.63$ counts) of 67 tTG-antibody negative samples (as defined by FEIA). For the two anti-tTG IgA-/IgG+ samples (marked with arrows), IgG depletion reduces the LFRET signal by 1) more than 59% and 2) below the positivity cutoff. For the IgA-positive and -negative samples (all samples not marked with arrows), a similar phenomenon is not seen.

Interestingly, IgG depletion reduced the tTG-LFRET signal to a varying extent in most of the anti-tTG positive samples (Fig 4). This suggests that not only anti-tTG IgA, but also anti-tTG IgG contributes to the tTG-LFRET signal. Anti-tTG IgG contribution to tTG-LFRET signal in samples with low concentration of anti-tTG IgA could also explain the enhanced sensitivity of tTG-LFRET for CD diagnosis without IgG depletion (87.8% vs 77.0% with IgG depletion, Figs 1 and 2). In the study population, this does not decrease the specificity of the method for CD diagnosis. This is in line with previous studies showing that anti-tTG IgG has a high specificity for CD comparable to anti-tTG IgA, albeit lower sensitivity [23].

In conclusion, our study shows that the LFRET approach is applicable to detection of IgA autoantibodies and serological diagnostics of autoimmune diseases. The assay appears highly specific in detection of CD, yet somewhat less sensitive. A homogenous approach, the LFRET is considerably faster than EIA, paralleling LFAs in both time and diagnostic performance. Unlike the LFA used, the LFRET provides a numeric result.

## Supporting information

**S1 Fig. Sensitivity and specificity (y-axis) of tTG-LFRET (tissue transglutaminase protein L TR-FRET assay) for CD (celiac disease) at different incubation times (x-axis).** We chose 22 minutes as the incubation time to achieve the best balance between sensitivity, specificity and assay time.
(TIF)

**S1 Table. Sensitivity and specificity of tTG-LFRET (tissue transglutaminase protein L TR-FRET assay) for CD (celiac disease) at different incubation times.** We chose 22 minutes (bolded) as the incubation time to achieve the best balance between sensitivity, specificity and assay time.
(DOCX)

## Acknowledgments

Lea Hedman is acknowledged for skillful technical assistance.

## Author Contributions

**Conceptualization:** Juuso Rusanen, Anne Toivonen, Jussi Hepojoki, Klaus Hedman.

**Data curation:** Juuso Rusanen, Anne Toivonen.

**Formal analysis:** Juuso Rusanen.

**Funding acquisition:** Jussi Hepojoki, Jorma Ilonen, Klaus Hedman.

**Investigation:** Juuso Rusanen, Jussi Hepojoki, Pekka Arikoski, Markku Heikkinen.

**Methodology:** Juuso Rusanen, Jussi Hepojoki, Satu Hepojoki.

**Project administration:** Jorma Ilonen, Klaus Hedman.

**Resources:** Anne Toivonen, Jussi Hepojoki, Outi Vaarala, Jorma Ilonen.

**Supervision:** Jussi Hepojoki, Jorma Ilonen, Klaus Hedman.

**Visualization:** Juuso Rusanen.

**Writing – original draft:** Juuso Rusanen, Anne Toivonen, Jussi Hepojoki.

**Writing – review & editing:** Juuso Rusanen, Anne Toivonen, Jussi Hepojoki, Satu Hepojoki, Pekka Arikoski, Markku Heikkinen, Outi Vaarala, Jorma Ilonen, Klaus Hedman.

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
