## [Decision Letter · Decision Letter 0]

8 Oct 2019

PONE-D-19-23614

LFRET, a novel rapid assay for anti-tissue transglutaminase antibody detection

PLOS ONE

Dear Dr. Juuso Rusanen:

Thank you for submitting your manuscript to PLOS ONE. After careful consideration, we feel that it has merit but does not fully meet PLOS ONE’s publication criteria as it currently stands. Therefore, we invite you to submit a revised version of the manuscript that addresses the points raised during the review process.

We would appreciate receiving your revised manuscript by November 2nd. To enhance the reproducibility of your results, we recommend that if applicable you deposit your laboratory protocols in protocols.io, where a protocol can be assigned its own identifier (DOI) such that it can be cited independently in the future. For instructions see: http://journals.plos.org/plosone/s/submission-guidelines#loc-laboratory-protocols

We look forward to receiving your revised manuscript.

Kind regards,

Sabato D'Auria

Academic Editor

PLOS ONE

Journal Requirements:

1. We note that you have included the phrase “data not shown” in your manuscript. Unfortunately, this does not meet our data sharing requirements. PLOS does not permit references to inaccessible data. We require that authors provide all relevant data within the paper, Supporting Information files, or in an acceptable, public repository. Please add a citation to support this phrase or upload the data that corresponds with these findings to a stable repository (such as Figshare or Dryad) and provide and URLs, DOIs, or accession numbers that may be used to access these data. Or, if the data are not a core part of the research being presented in your study, we ask that you remove the phrase that refers to these data.

Reviewers' comments:

Reviewer's Responses to Questions

**Comments to the Author**

1. Is the manuscript technically sound, and do the data support the conclusions?

Reviewer #1: Yes

Reviewer #2: Yes

2. Has the statistical analysis been performed appropriately and rigorously? 

Reviewer #1: Yes

Reviewer #2: I Don't Know

3. Have the authors made all data underlying the findings in their manuscript fully available?

Reviewer #1: Yes

Reviewer #2: Yes

4. Is the manuscript presented in an intelligible fashion and written in standard English?

Reviewer #1: Yes

Reviewer #2: Yes

5. Review Comments to the Author

Reviewer #1: In the manuscript “LFRET, a novel rapid assay for anti-tissue transglutaminase antibody detection”, the authors present the application of previous L-based time-resolved Foster energy transfer (TR-FRET) assay to detect anti-tissue transglutaminase antibody in patients.

In my opinion, this work is of interest for the readers of the Journal but I recommend the publication after a minor revisions. In particular:

- The application of TR-FRET assay from the authors and the comparing the sensitivity and specificity with other available test reported in this manuscript appears to be very promising to shed light on the individuation of point-of-care test. I know that in previous work the authors have described in details the approach, but could be an add value if they integrate in this manuscript a flow chart taht describe the assay. This will allow the reader to better understand the assay. Please the authors provide it.

- In the materials and methods section the paragraph “Reference methods” and “Statistically analysis” should be unified. Please, the authors provide it.

-Materials methods section the paragraph “tTG-LFRET assay” and “IgG depletion” could be unified. Please, the authors consider this option.

- In the paragraph “Sample” of the materials and method section, please add more information about the sample preparation before to perform the assays.

- All figure legends present in the manuscript lack of important details that allow the reader to understand the single figures. Please, the authors improve it.

- Please at line 48 clarify the abbreviation HLA

Reviewer #2: In the present paper, Dr. Juuso Rusanen and co-workers exploited a recombinant Eu-labeled tTG antigen and AF-labeled protein L to develop a TR-FRET-based immunoassay (named LFRET) for the detection of anti-tTG antibodies in celiac disease patients.

They validated the assay on serum samples form 74 celiac patients and 70 healthy subjects, finding a good correlation between the LFRET assay and current immunoassays (both fluorescence enzyme immunoassay (FEIA) and lateral flow assay (LFA)).

The new LFRET assay is conceived as a point of care test, with the advantage of being quantitative, compared to LFA. Despite the data are quite solid, certain aspects are not clear and need to be revised.

Specific comments:

1) In the methods section a description of the LFRET assay is necessary; the authors should at least mention how serum/plasma samples are processed, how the assay is performed, and what instrument they use for fluorescence acquisition and analysis.

2) It is not clear how the FRET signal is shown (figure 1 and figure 2). The authors state that “ LFRET signal is expressed as average of normalized acceptor wavelenght emission counts from two replicats of the same sample (with 2 consecutive measurements from both replicates)”. But, since counts are “not divided by the TR-FRET signal of the negative control”, what normalization has been done? On the x-axis “LFRET signal, counts, log scale” is reported, but as I can understand the scale is not in log10? Reported numbers are referred to counts x103? The rationale for the cut off value and the Pearson correlation coefficient should be reported in the figure legend.

3) In figure 2 the authors report the results of the LFRET assay after IgG depletion, correlating them with results of the FEIA assay (that is specific for IgA). It is not clear how the cutoff for FEIA is calculated. In addition, no correlation coefficient is reported.

4) In figure 3 the LFRET counts of the samples before and after IgG depletion are shown, in order to demonstrate that the new system is able to detect anti-tTG IgA-negative yet IgG-positive samples. A threshold for identifing these samples was established, and validated on two samples taken from the HUSLAB bank (laboratory), outside of the sample collection of the study. In practice, these samples serve as controls to prove the validity of the treshold, but this point should be explained more clearly in the text. In addition, also for figure 3 it is not clear how the cutoff of counts was calculated.

5) The first results section is intitled tTG-LFRET incubation time, cutoff and performance but no data are shown about the incubation time. The title should be changed.

6) Since the results from FEIA and from LFA are reported in the second paragraph I suggest to move the last sentence of the first section (line 152-154, and the discussion of the figure 1) to the second section. Also, the title of the second paragraph should indicate a comparison between the reference methods and the new one. Table 2 should be commented in this results section.

7) In table 2 it should be more correct to report mean ±SD of the duplicates samples for LFRET and FEIA.

Minor points

1) Use always the same acronim for FEIA (or EIA)

2) Table 1 is subdivided in three subtables that should be combined in a unique one.

3) How are considered the samples between 7 and 10 U/ml in the FEIA assay? This should be specified in the methods/results interpretation.

4) Raw 230: insert the reference as a number

6. PLOS authors have the option to publish the peer review history of their article (what does this mean?). If published, this will include your full peer review and any attached files.

Reviewer #1: No

Reviewer #2: No

---

## [Author Response · Author response to Decision Letter 0]

2 Nov 2019

Response to Reviewers Juuso Rusanen 2.11.2019

Dear Dr. Juuso Rusanen:

Thank you for submitting your manuscript to PLOS ONE. After careful consideration, we feel that it has merit but does not fully meet PLOS ONE’s publication criteria as it currently stands. Therefore, we invite you to submit a revised version of the manuscript that addresses the points raised during the review process.

[…]

Kind regards,

Sabato D'Auria

Academic Editor

PLOS ONE

-> Our warmest thanks for the opportunity to revise our manuscript according to journal requirements and reviewers’ comments, for which point-by-point replies are included below.

Journal Requirements

[…]

1. We note that you have included the phrase “data not shown” in your manuscript. Unfortunately, this does not meet our data sharing requirements. PLOS does not permit references to inaccessible data. We require that authors provide all relevant data within the paper, Supporting Information files, or in an acceptable, public repository. Please add a citation to support this phrase or upload the data that corresponds with these findings to a stable repository (such as Figshare or Dryad) and provide and URLs, DOIs, or accession numbers that may be used to access these data. Or, if the data are not a core part of the research being presented in your study, we ask that you remove the phrase that refers to these data.

-> Thank you for the remark and our sincere apologies for not noticing this when first submitting the manuscript. We have now added supporting information files including the data referred to in the part of the manuscript in question.

Reviewer #1

In the manuscript “LFRET, a novel rapid assay for anti-tissue transglutaminase antibody detection”, the authors present the application of previous L-based time-resolved Foster energy transfer (TR-FRET) assay to detect anti-tissue transglutaminase antibody in patients.

In my opinion, this work is of interest for the readers of the Journal but I recommend the publication after a minor revisions. In particular:

- The application of TR-FRET assay from the authors and the comparing the sensitivity and specificity with other available test reported in this manuscript appears to be very promising to shed light on the individuation of point-of-care test. I know that in previous work the authors have described in details the approach, but could be an add value if they integrate in this manuscript a flow chart taht describe the assay. This will allow the reader to better understand the assay. Please the authors provide it.

-> We are very happy to hear that the reviewer considers the manuscript very promising, and agree that a flowchart would help better understand the assay. Thus, we have added one (Fig 1).

- In the materials and methods section the paragraph “Reference methods” and “Statistically analysis” should be unified. Please, the authors provide it.

-> We have unified the sections accordingly.

-Materials methods section the paragraph “tTG-LFRET assay” and “IgG depletion” could be unified. Please, the authors consider this option.

-> We agree with the reviewer that unifying these sections would make sense, and tried this option. However, we found that unifying these sections and placing the second paragraph (comparison of methods) as last would perhaps disturb the flow of the manuscript more than the current order of paragraphs, which we thus chose to keep. For clarity, we have renamed the section ”IgG depletion” to ”IgG depletion in tTG-LFRET”. 

- In the paragraph “Sample” of the materials and method section, please add more information about the sample preparation before to perform the assays.

-> We have added a flowchart (Fig 1), which also describes the sample preparation.

- All figure legends present in the manuscript lack of important details that allow the reader to understand the single figures. Please, the authors improve it.

-> We fully agree, and therefore have added significantly more information to figure legends, including but not limited to clarifying all non-standard abbreviations used.

- Please at line 48 clarify the abbreviation HLA

-> We have clarified this abbreviation.

Reviewer #2

In the present paper, Dr. Juuso Rusanen and co-workers exploited a recombinant Eu-labeled tTG antigen and AF-labeled protein L to develop a TR-FRET-based immunoassay (named LFRET) for the detection of anti-tTG antibodies in celiac disease patients.

They validated the assay on serum samples form 74 celiac patients and 70 healthy subjects, finding a good correlation between the LFRET assay and current immunoassays (both fluorescence enzyme immunoassay (FEIA) and lateral flow assay (LFA)).

The new LFRET assay is conceived as a point of care test, with the advantage of being quantitative, compared to LFA. Despite the data are quite solid, certain aspects are not clear and need to be revised.

-> Thank you.

Specific comments:

1) In the methods section a description of the LFRET assay is necessary; the authors should at least mention how serum/plasma samples are processed, how the assay is performed, and what instrument they use for fluorescence acquisition and analysis. 

-> We agree with comment and have added a flowchart illustrating how the samples are processed and the assay performed (Fig 1). Furthermore, the instrument used for fluorescence measurements has been named in the methods section.

2) It is not clear how the FRET signal is shown (figure 1 and figure 2). The authors state that “ LFRET signal is expressed as average of normalized acceptor wavelenght emission counts from two replicats of the same sample (with 2 consecutive measurements from both replicates)”. But, since counts are “not divided by the TR-FRET signal of the negative control”, what normalization has been done? On the x-axis “LFRET signal, counts, log scale” is reported, but as I can understand the scale is not in log10? Reported numbers are referred to counts x103? The rationale for the cut off value and the Pearson correlation coefficient should be reported in the figure legend.

-> We appreciate the points raised by reviewer #2. For clarification, we have added a reference regarding normalization to this paper to the materials and methods section, line 131. To further clarify, normalization here refers to normalizing the acceptor (Alexa Fluor 647, AF647) signal measured at 665 nm with respect to the donor (Europium) emission at 665 nm. This is done according to the following equation (from Saraheimo et al. 2013): AF647N = AF647–k*Eu, where AF647N = normalized AF647 fluorescent counts, AF647 = unnormalized A647 counts (at 665 nm), k = Eu emission at 665 nm/Eu emission at 615 nm and Eu = Eu fluorescent counts (at 615 nm).

The x axis scale is logarithmic, but the axis labels are indeed arbitrarily set to best cover the signal area (25-300). Alternatively, we could use e.g. log2 axis labels, but in our opinion this would make the figure more difficult to perceive (see Figs i & ii in the file "Response to Reviewers"). To clarify this part, we now mention in the axis titles that axis labels are arbitrary.

We have added the rationale to the cutoff as well as Pearson correlation coefficients to the figure legends (Figs 2 and 3).

3) In figure 2 the authors report the results of the LFRET assay after IgG depletion, correlating them with results of the FEIA assay (that is specific for IgA). It is not clear how the cutoff for FEIA is calculated. In addition, no correlation coefficient is reported.

-> We have added Pearson correlation coefficient to the figure legend. The cutoff of FEIA assay is manufacturer-defined.

4) In figure 3 the LFRET counts of the samples before and after IgG depletion are shown, in order to demonstrate that the new system is able to detect anti-tTG IgA-negative yet IgG-positive samples. A threshold for identifing these samples was established, and validated on two samples taken from the HUSLAB bank (laboratory), outside of the sample collection of the study. In practice, these samples serve as controls to prove the validity of the treshold, but this point should be explained more clearly in the text. In addition, also for figure 3 it is not clear how the cutoff of counts was calculated.

-> Thank you for these helpful remarks, we have clarified this point in the text accordingly. Furthermore, we now explain the algorithm for differentiating between IgG+/IgA- and IgA+ samples in the figure legend.

5) The first results section is intitled tTG-LFRET incubation time, cutoff and performance but no data are shown about the incubation time. The title should be changed.

-> We now have added data on incubation time (S1 Fig and S1 Table in Supplementary data).

6) Since the results from FEIA and from LFA are reported in the second paragraph I suggest to move the last sentence of the first section (line 152-154, and the discussion of the figure 1) to the second section. Also, the title of the second paragraph should indicate a comparison between the reference methods and the new one. Table 2 should be commented in this results section.

-> Thank you, we have amended the manuscript accordingly.

7) In table 2 it should be more correct to report mean ±SD of the duplicates samples for LFRET and FEIA.

-> We now have corrected the table to refer to means ± SD of duplicates for LFRET. For FEIA, no duplicate measurements were taken.

Minor points

1) Use always the same acronim for FEIA (or EIA)

-> In the manuscript, we refer to EIA as a more general concept in the abstract, introduction and conclusions, as also non-fluorescent EIAs are used for anti-tTG antibody determination. We use the word FEIA only when specifically referring to the reference EIA we used in this study (which is a fluorescent EIA, or FEIA). We hope that this adequately explains why we consider it justified to use the acronyms separately.

2) Table 1 is subdivided in three subtables that should be combined in a unique one.

-> Thank you for the excellent suggestion, we have combined the three subtables into a single table.

3) How are considered the samples between 7 and 10 U/ml in the FEIA assay? This should be specified in the methods/results interpretation.

-> In this manuscript, results equivocal in FEIA (between 7 and 10 U/ml) are considered positive. There were two samples of this range, one true and one false positive. We have added a mention of this on page 8, line 179.

4) Raw 230: insert the reference as a number

-> We now have added the reference.

---

## [Editor Report · Decision Letter 1]

14 Nov 2019

LFRET, a novel rapid assay for anti-tissue transglutaminase antibody detection

PONE-D-19-23614R1

Dear Dr. Juuso Rusanen,

We are pleased to inform you that your manuscript has been judged scientifically suitable for publication and will be formally accepted for publication once it complies with all outstanding technical requirements.

With kind regards,

Sabato D'Auria

Academic Editor

PLOS ONE
---

## [Editor Report · Acceptance letter]

19 Nov 2019

PONE-D-19-23614R1 

LFRET, a novel rapid assay for anti-tissue transglutaminase antibody detection 

Dear Dr. Rusanen:

I am pleased to inform you that your manuscript has been deemed suitable for publication in PLOS ONE. Congratulations! Your manuscript is now with our production department. 

With kind regards,

on behalf of

Dr. Sabato D'Auria 

Academic Editor

PLOS ONE